# Sex-Related Differences in Pancreatic Ductal Adenocarcinoma Progression and Response to Therapy

**DOI:** 10.3390/ijms252312669

**Published:** 2024-11-26

**Authors:** Jelena Grahovac, Ana Đurić, Miljana Tanić, Ana Krivokuća

**Affiliations:** Experimental Oncology Department, Institute for Oncology and Radiology of Serbia, Pasterova 14, 11000 Belgrade, Serbia

**Keywords:** PDAC, sex disparity, tumor immunity, therapy

## Abstract

Pancreatic ductal adenocarcinoma (PDAC) is one of the most deadly malignancies with an increasing incidence rate and limited therapeutic options. Biological sex has an impact on many aspects of PDAC development and response to therapy, yet it is highly unappreciated in both basic and translational research, and worryingly in PDAC clinical trials. In this review, we summarize how biological sex influences PDAC incidence and mortality, genetic and epigenetic landscapes, anti-tumor immunity, responses to hormones, cachexia, and the efficacy of therapy. We highlight the importance of sex as a variable and discuss how to implement it into preclinical and clinical research. These considerations should be of use to researchers aiming at improving understanding of PDAC biology and developing precision medicine therapeutic strategies.

## 1. Introduction

Pancreatic ductal adenocarcinoma (PDAC) is one of the most lethal cancers in the world with an increasing incidence rate. As no effective screening tests are available, more than 80 percent of PDAC patients have inoperable or metastatic disease at the time of diagnosis. This precludes curable resection in the majority of patients. Borderline resectable and locally advanced PDAC patients undergo neoadjuvant chemotherapy ± radiation and exploration, while metastatic patients receive chemotherapy. PDAC is highly resistant to conventional chemotherapy, due to the presence of inflammatory and fibrotic stroma, hypovascularization, intra- and inter-tumor heterogeneity, and the ability to metabolically adapt to changes in the microenvironment. The checkpoint inhibitor blockade has also been ineffective in several clinical trials in PDAC [1,2,3]. With advancements in the understanding of the PDAC pathophysiology in the past two decades, the relative 5-year survival rate has increased from 4 percent to 13 percent in the United States [4] and to 8 percent in Europe [5]. Still, pancreatic cancer is the third leading cause of cancer-related death in the United States and the fourth in Europe, and novel approaches and treatments for PDAC are urgently needed. There is growing evidence that in non-reproductive cancers biological sex has influence on the incidence and mortality [6], cancer metabolism, tumor immune response and the patient response to therapy [7,8]. In PDAC, both the estimated number of new cases as well as the estimated number of deaths are higher in males than in females [4,9], yet sex is not considered in treatment decision-making. A meta-analysis of 25 studies including more than 27,000 patients revealed that female sex was significantly associated with long-term survival (>5 years) after PDAC resection [10]. However, knowledge on the impact of sex on the development of PDAC is limited, as well as the understanding of the sex-related differences in response to treatment. In this review, we summarize the biology behind the sex differences and the evidence of sex disparity from preclinical and clinical research, and propose measures for incorporating segregated sex analysis in research to better tailor novel treatment approaches.

## 2. Sex and Gender Defined

Both the biological and environmental differences between males and females are contributing to the observed sex disparity in cancer. Many researchers from the biomedical field are still unfamiliar with the distinction between sex and gender and often use these terms interchangeably. In humans, biological sex refers to two sexes—male and female—determined by sex chromosomes, hormones and reproductive organs. The examination of the impact of sex on human health requires an integrative approach, as it is dynamic and changes throughout lifetime [11]. Important sex differences between males and females exist in immune response, body composition (percent of fat and muscle), hormonal status, metabolism, and the pharmacokinetics and pharmacodynamics of drugs, and they can all have an impact on the development of cancer and response to therapy. Gender is a sociocultural construct which takes into account attitudes, behaviors, and identities, and can change with time and place, varies between societies, and intersects with biological sex, sexual orientation, age, socioeconomic status, and ethnicity [12]. The impact of sex can be studied in the preclinical setting, in vitro and in vivo, but gender can only be taken into account as a variable in human clinical trials and cannot be modeled in the biomedical laboratory setting. Sex and gender are highly correlated, making sex a confounding factor in studies that examine the relationship between lifestyle- and/or occupational-related exposures and outcomes. In genetic epidemiology studies, gender can be considered as a compound or surrogate marker for a complex interplay between sex and various socio-economic, behavioral, and cultural factors that affect health outcomes in pancreatic cancer. While the effects of biological sex are the same irrespective of the population, and thus generalizable, the effects of gender are not, as they depend on the cultural norms in different societies. To discern the true effects of sex, it is not enough to use matched groups or to control for (adjust) sex or gender in a study. Since biological sex can modify the effects of a variable of interest on a health outcome, a stratified analysis is necessary to uncover this interaction [13].

In this review, male (men) and female (women) will refer to the biological sex assigned at birth.

## 3. Disparity in PDAC Incidence and Survival

The risk of developing any gastrointestinal cancer is higher in men, than in women [11]. Based on the data from the International Agency for Research on Cancer (https://gco.iarc.fr/today/en, accessed on 8 September 2024) and the Surveillance, Epidemiology and End Results (SEER) Program from the National Cancer Institute (https://seer.cancer.gov/registries/terms.html, accessed on 8 September 2024), the age-standardized incidence for pancreatic cancer is higher in males than in females, while the highest lifetime risk for developing pancreatic cancer is reported for Western Europe [14]. In the past two decades, pancreatic cancer incidence increased for both sexes (Figure 1A), but the relative increase in incidence followed the age- and sex-specific trend.

Namely, in the group aged 55 years or older, there was a greater increase among the men, while in the group younger than 55 years, there was a significantly greater relative increase in incidence occurred among the women [15]. Especially worrisome was the increased incidence rate of pancreatic cancer in the young females (post hoc analysis in group aged 15 to 34 years from the SEER data released in 2021 [15] or post hoc analysis in group aged 18 to 26 years from the SEER data released in 2022 [16]), compared to the males of the same age group (SEER data released in 2023, represented in Figure 1B).

The PDAC age standardized mortality rate is higher in men than in women, but when stratified in age groups, the effect size is different. The highest differences in the relative survival was present in the age group under 44 years (19.25% in males and 27.33% in females) while the difference was lost in the age group older than 65 years (https://ecis.jrc.ec.europa.eu/explorer.php, accessed on 8 September 2024.). Based on the SEER data, women have a higher 5-year relative survival rate in the age group younger than 50 years, with a smaller advantage in the group 50–64 years of age, while in the >65 group, the advantage was lost (Figure 1C). The five-year relative survival rate in the age group < 50 was better in the females across patients with the localized, regionally spread cancer and the patients with distant PDAC metastases. In the meta-analysis of data from the International Pancreatic Cancer Case–Control Consortium [17], the women had higher numbers of early diagnoses in stage I–II, and a lower number of stage III–IV diagnoses than the men in the five analyzed countries. Unfortunately, even the large population-based studies looking into the survival estimates based on age or resection status often do not report sex-stratified analysis [18]. Exposure to carcinogens, a smoking habit or alcohol consumption, care seeking and therapeutic choices are gender based, but even after accounting for the environmental effects, men still showed higher tumor incidence than women in most cancer types. This implies that in cancer development and response to treatment there are fundamental biological differences between males and females [6]. It is also important to note that the available incidence and survival data were mostly based on the data from high-income countries, as low- and middle-income countries do not have quality cancer registries [19,20,21].

## 4. Genetic Differences Between Sexes in PDAC

More than 90% of patients with PDAC carry a *KRAS* mutation [22], followed by *TP53*, *CDKN2A*, and *SMAD4* [23]. In the targeted sequencing of 2336 PDAC samples and their matched healthy tissues via the MSK-IMPACT platform (https://www.cbioportal.org/study/summary?id=pdac_msk_2024, accessed on 10 September 2024), no differences were found between the males and females in the frequency of the mutation in the 10 most commonly mutated genes in PDAC (Figure 2A). However, numerous genes that engage in p53 networks are located on the X chromosome [24], and the presence of two X chromosomes offers unique cancer protection for females, across tumor types [25]. The differential expression of tumor suppressor genes located on the X-chromosome pseudo-autosomal regions (PAR) or those that escape X-inactivation (EXITS), render males more susceptible to the functional loss of an X-linked TSG [26]. For example, a bi-allelic loss of histone lysine demethylase *KDM6A* leads to squamous-like PDAC in females [27], while in males, mutations in *KDM6A* were more often associated with the loss of its Y-chromosome homologue *UTY* [26,27].

The polygenic risk scores (PRS) identified through genome-wide association studies (GWAS) have been useful to quantify genetic susceptibility to pancreatic cancer mediated by a combination of common genetic polymorphisms (SNPs) with individually small effects [28,29]. Unfortunately, while many studies looking into genetic and environmental risk factors for PDAC adjusted for sex and known risk factors, all but one did not report analysis results stratified by sex [30,31,32,33,34,35]. A great example of a properly conducted study looking into the effects of sex and gender-related lifestyle and socioeconomic factors on genetic risk for PDAC is by Zeng et al. [36]. The authors disaggregated a cohort of 340,631 subjects from the UK Biobank according to the polygenic risk score (PRS), composite lifestyle score and reported sex, to uncover the stronger association of polygenic risk scores with the PDAC risk in females (HR 1.55; 95% CI, 1.26–1.91) and the stronger protective association of favorable (healthy) lifestyle in males (HR 0.46; 95% CI, 0.38–0.57). The highest risk of developing PDAC was observed for males with high polygenic risk scores and unfavorable lifestyles. Another study that looked at the single *TP53* Arg 72 Pro polymorphism, which has been reported as a risk factor in several cancer types, found significantly strong association with PDAC risk only in males, but not in females [37]. The association was particularly strong in heavy smokers and excessive alcohol drinkers.

Across tumor types, greater somatic mutational load was observed in the males compared to the females, as well as differences in mutational signatures that could be explained by environmental, thus gender-related and/or sex-based, determinants [38]. Strikingly, X and Y chromosomes themselves were most commonly excluded from the analysis.

A seminal study by Yuan et al., on the molecular-level differences between male and female cancer patients in 13 cancer types, revealed both sex-biased gene expression patterns for more than 60 individual genes, and sex-biased molecular signatures comprising more than a thousand genes per signature [39]. Unfortunately, PDAC was not included in the analyzed cancer types, and these differences remain to be explored. Importantly, the Yuan et al. study found that more than 50 percent of clinically actionable genes showed sex-biased expression.

## 5. Methylation Differences Between Sexes in PDAC

Human methylome is affected by both genetic and environmental factors, whose impact on thousands of CpG sites varies between males and females, and across lifespans [40,41]. These include metabolite-associated CpGs (tryptophan levels, manose, and 5-oxoproline) that display sex difference in the impact of environmental factors modulating their methylation levels [40]. Environmental influences on the epigenome encompass many types of exposures, such as nutrition, chemicals/pollutants, stress, and others, that are often gender-related and are hard to disentangle from sex-based effects.

Most of the epigenome-wide association studies (EWAS) in PDAC use peripheral blood as a surrogate tissue [42,43]. A study performed using blood collected >10 years from the pre-diagnosis of PDAC in 393 cases and 431 controls, found differential methylation between the cases and healthy controls in the promoter region of the genes *TMEM204* and *IFT140*, *MFSD6L*, *FAM134B/RETREG1*, *KCNQ1D*, and *C6orf227*, and highlighted the alterations in the gastric secretion pathway [44]. Unfortunately, although the authors performed a stratified analysis for smoking and BMI, showing a stronger association among the current smokers and the overweight/obese cases, sex was not adjusted for nor used for stratification. A more recent 2-phase EWAS, using pre-diagnostic blood from 44 cases and 556 controls, and 13 cases and 26 controls, respectively, validated six CpGs (in or near *AIM2*, *DGKA*, *STK39*, and *TNFSF8*) and three differentially methylated regions (DMRs) in or near the *nc886*, *LY6G5C*, and *HLA-DPB1* gene promoters, suggesting, through Mendelian randomization (MR) analysis, the causal effect of *nc886* and *STK39* gene methylation on PDAC susceptibility [45]. Sex was adjusted for in the first cohort, while the second cohort was female-only. A commendable example where the authors both adjusted their model for age and sex, and performed a stratified analysis according to sex and lifestyle factors, is the study where the hypermethylation of LINE-1 repetitive elements in the peripheral blood of the PDAC patients compared to the healthy controls was associated with a greater risk of PDAC [46]. However, no interaction between the sex and LINE-1 methylation levels was found. Increased levels of DNA hydroxymethylation (5-hm) were observed in the enhancer region of the PDAC driver genes such as *MYC*, *KRAS*, *VEGFA*, and *BRD4* [43] in the tumor tissue compared to normal, and in cfDNA; however, despite the sex-specific hydroxymethylation observed in neuroendocrine pancreatic cancer [47], none of these studies examined the effect of sex. An inspection of the TCGA Firehouse legacy PDAC Methylation HM450 dataset (https://www.cbioportal.org, accessed on 1 October 2024) revealed at least 25 significantly differentially methylated genes between males and females (Figure 2B), inviting further exploration.

## 6. Hereditary PDAC

Familial pancreatic cancer, defined by the occurrence of PDAC in two or more first-degree relatives without an identified genetic mutation, accounts for 4–10% of PDAC cases [48]. 5–10% of these PDAC cases are linked to known gene mutations, equating to about 0.6–1.3 cases per 100,000 people each year. Multiple genetic syndromes increase susceptibility to PDAC, including mutations in genes such as *BRCA1*, *BRCA2*, and *PRSS1*, and those involved in mismatch repair [49]. Lynch syndrome, for instance, is linked to a rare “medullary” subtype of PDAC. Additionally, PDAC has been associated with other hereditary conditions, such as hereditary breast and ovarian cancer, hereditary pancreatitis, Peutz–Jeghers syndrome, and familial atypical multiple-mole melanoma syndrome. While the percentage of hereditary PDAC (hPDAC) cases is relatively small, recognizing the genetic risk is essential for early detection and prevention [48]. In addition, germline genetic testing plays a crucial role in cancer treatment, as *BRCA1/2* mutations are predictive of a positive response to PARP inhibitors and platinum-based chemotherapy, while mutations in mismatch repair genes suggest potential benefits from immune checkpoint inhibitors [50]. Germline mutations are found in about 10% of patients with PDAC [51] with the most frequent mutations occurring in *BRCA1/2*, *ATM*, and MMR genes [52]. These mutations are more commonly observed in younger patients under 60, with no significant differences in mutation prevalence between the males and females in sporadic forms. Due to the prevalence and clinical relevance of these mutations, genetic counseling and testing are recommended for all PDAC patients, irrespective of age, sex, or family history [51].

Research into sex-based differences in hPDAC is still emerging, with some studies investigating how genetic predispositions and outcomes vary by sex. A recent multicenter study in Taiwan revealed that a greater proportion of patients with hereditary predispositions to PDAC were male, mirroring the higher overall incidence of pancreatic cancer in men [53]. Certain germline mutations, such as those in *BRCA1* and *BRCA2*, have also shown sex-related differences in frequency and impact. *BRCA2* mutations, for example, are more commonly linked to an elevated risk of pancreatic cancer in men, as well as to other cancers like prostate cancer and melanoma. Although *BRCA1* mutations are less common in males, they still contribute significantly to the risk of pancreatic and other cancers. Approximately 19% of all *BRCA* mutation carriers with pancreatic cancer are male, while the remaining 81% are female [54]. Another systematic review also highlighted that while genetic mutations such as *BRCA1/2* are prevalent in both sexes, their clinical implications may vary by gender, potentially influencing screening and management approaches [55].

Unfortunately, studies reporting the survival of carrier PDAC patients treated with platinum-based therapy most commonly do not perform nor report a sex-segregated analysis of response [56]. A recent study compared the survival of PDAC patients with mutations in eight genes involved in HRR *(ATM*, *BARD1*, *BRCA1*, *BRCA2*, *BRIP1*, *PALB2*, *RAD51C*, and *RAD51D)* with patients testing negative for mutations [57]. While further exploratory analyses were performed by the stage of diagnosis, the year of diagnosis, and the type of chemotherapy, a sex-segregated analysis was not performed. The HRR mutation carriers were younger and more likely to have metastatic disease at diagnosis, while 60% of the patients in the carrier group were males.

Interestingly, in the individuals with hereditary pancreatitis that faced a significantly elevated risk of developing pancreatic cancer, with an estimated cumulative risk of 40% by the age of 70, the risk increased to 75% in the cases of paternal inheritance [58].

Most studies do not distinguish between the mortality rates of hereditary and sporadic PDAC, and hereditary PDAC shares the same poor prognosis as sporadic forms.

## 7. Immune Differences Between Sexes in PDAC

Cancer is, in part, a result of impaired immune surveillance where differences in anticancer immunity could contribute to a higher cancer incidence and a poorer outcome in men [59]. The ability of the host to control the tumor is better in females, as revealed by clinical outcomes including the PDAC patients [8]. Sex has an impact on both innate and adaptive immune response and is an important factor that influences the tumor microenvironment. The differential immune response between males and females is regulated by sex chromosomes, sex hormones, nutrition, and microbiota [60]. Females are more immunocompetent and have higher leukocyte numbers than males [61]. In healthy individuals, differences in lymphocyte subsets, including B cells (higher in females), CD4+ T cells (higher in females), CD8+ T cells (higher in males), and CD4/CD8 ratios (higher in females) are well documented [60]. When analyzed by sex, in gene set enrichment analyses (GSEAs) tumor immune signatures differ between males and females in many cancer types, including PDAC [8].

Many genes that are involved in adaptive and innate immunity are X-linked [62]. Three X-linked genes that can escape X-inactivation and have a role in immune response—*FOXP3* [63], *TLR7* and *CD40L* [64]—are associated with increased susceptibility to autoimmune disease in females, but may present an advantage in anti-tumor response. Based on the analysis of both cancer tissues and blood samples, it was found that in PDAC, females have a stronger systemic immune response compared to males [65]. In mouse models of the disease, male mice had accelerated PDAC tumor progression, and these differences were not observed in immunodeficient mice [66]. It was found that androgen signaling suppressed T cell immunity against cancer in males, and that reduction in testosterone synthesis via surgical castration or using the small-molecular inhibitor abiraterone significantly enhanced the antitumor activity of T cells in male mice and improved the efficacy of anti–PD-1 immunotherapy [66]. It was also reported that mutated *KRAS* itself drives immune evasion in the genetically engineered mouse model (GEMM) of PDAC [67]. In this study, it was shown that the KRAS-deficient cells triggered a strong antitumor response. Unfortunately, while the authors used both male and female mice (8–10 weeks old), that were housed individually; they did not analyze the results of the study based on sex.

Recently, using the analytical framework for the multiparametric integration of multi-layered bioimaging datasets, Aliar et al. uncovered a so-far unknown sexual dimorphism in the IL-6/STAT3-linked intratumoral T cell response in human pancreatic cancer [68]. In this study, it was pointed out that there was a male-specific regional T cell response linked to IL-6 levels. It was concluded that the tumor biological effects of IL-6/STAT3 signaling were similar between the PDAC subtypes, but differed strongly between the sexes, especially with regard to T cell populations. This is of importance, as the previous animal studies that described tumor-stroma IL-6/STAT3 communication [69] and reported the efficacy of the combined STAT3 and MEK inhibition to overcome immunotherapy resistance in PDAC [70,71] did not report the sex of the animals used. The study of McAndrews et al., looking at the distinct IL-6-mediated therapy resistance in PDAC, used both male and female mice with various GEMM backgrounds, but the results were not reported by sex, nor was the expression of cancer-associated fibroblast (CAF) markers and cytokines that were studied reported by sex [72]. Given the insights into the sexual disparity in cachexia (discussed further in the text) and the tumor immune infiltration related to IL-6 signaling, it will be important to pay attention to the sex-segregated analysis of IL-6/STAT3 modulation in PDAC, especially as there is an ongoing clinical trial for combined MEK, STAT3, and PD-1 inhibition in metastatic PDAC (NCT05440942).

Taken together, immune responses in the tumors in males and females should be studied separately, using both female- and male-derived model systems in vitro, animals of both sexes in vivo, and patient selection strategies in clinical studies that will better hone immunotherapy approaches.

## 8. The Role of Sex Hormones in PDAC

Male sex hormones, androgens, are steroids synthetized in the gonads and the adrenal gland. Androgen receptors are expressed on T cells, B cells, macrophages, monocytes, and neutrophils, and the androgen-mediated suppression of immune reactivity is thought to be one of the sex-specific biases in immunity [73]. In the rat model of pancreatic cancer, sex steroids had a major role in the higher incidence in male versus female rats [74]. Namely, azaserine-induced pancreatic carcinogenesis preneoplastic lesions were more frequent in male than in female rats, and the frequency of lesions was increased in the ovariectomized or testosterone-treated females, and decreased in the orchiectomized males compared to the intact animals.

In females, the follicle-stimulating hormone (FSH), the luteinizing hormone (LH), the estradiol, and the progesterone present in systemic circulation influence all the organs in the body. It has been reported that multiple pregnancies and the use of estradiol-containing oral contraceptives were associated with a decreased PDAC risk [75,76,77]. In the prospective, population-based cohort of more than 17,000 women in the Malmö Diet and Cancer Study, a higher age at menarche was significantly associated with increased pancreatic cancer risk (age-adjusted HR = 1.17) and the use of hormone replacement therapy (HRT) was significantly associated with a decreased risk of pancreatic cancer (age-adjusted HR = 0.47), in particular, the use of estrogen-only regimen (age-adjusted HR = 0.21) [77].

Estrogens exert their effects through estrogen α (ERα) and β receptors (ERβ) and the G protein-coupled estrogen receptor (GPER). Estrogen receptors exist in normal and neoplastic human pancreatic tissue [78]. PDAC cells express estrogen receptors, and the effects of the variety of chemotherapeutic drugs in in vitro tests (5-fluorouracil, cisplatin, docetaxel, etc.) were enhanced in the presence of β-estradiol [79], with the hypothesized mechanism being that low dose β-estradiol treatment stimulates cell cycle progression, which makes more of the cycling cancer cells sensitive to various drugs. It is postulated that hormonal fluctuations within the menstrual cycle phase may be a primary cause of documented sex differences in the pharmacokinetics and pharmacodynamics of drugs in patients [80].

In early studies with the estrogen receptor modulator tamoxifen, it was shown that it accumulates in the mouse pancreata [81]. Tamoxifen exhibits both estrogenic agonist and antagonist effects in different parts of the body, and is a GPER agonist. Non-classical estrogen signaling through the GPER may be tumor suppressive [82]. In patients with unresectable or incompletely resected PDAC, tamoxifen-treated patients had a prolonged survival benefit (7 vs. 3 months), and the effect was most prominent in women older than 60 years (12 months) [83]. In the mouse models of PDAC, including human xenografts, highly specific GPER agonist G-1-induced tumor regression significantly prolonged survival, and increased the efficacy of PD-1-targeted immune therapy [84]. Unfortunately, in this particular study, all the mice were female, and it was not discussed whether the effect was sex dependent. In another study showing the GPER-mediated tamoxifen effects on the normalization of the PDAC stroma, the authors used mice of both sexes (2–3 males and 2–3 females per group), but the effects were not analyzed by sex [85].

The liver is the most frequent metastatic site in PDAC, and patients with liver metastases have worse outcomes compared to patients with lung or distant nodal metastases [86]. Liver is a sexually dimorphic organ in which estrogen and androgen receptors are present, and estrogens have been found to play a protective role [87]. This is in concordance with recent findings that male PDAC patients have more frequent liver metastasis, and elevated hepatic metastasis-promoting gene expression compared to females [88].

Taken together, the sex hormones present in systemic circulation can have an impact on both tumor development and the tumor response to therapy in PDAC, and a sex-segregated analysis is necessary in both pre-clinical and clinical settings.

## 9. Sex Disparities in PDAC Cachexia

PDAC is characterized by cachexia, an inflammation-related state with increased catabolism and skeletal muscle and fat waste [89]. Cancer-associated cachexia (CC) is characterized by weight loss, muscle atrophy, fatigue, and poor outcomes. The majority of patients with unresectable PDAC by the time of diagnosis have undergone significant weight loss [90], and up to 85% of all PDAC patients are affected by CC during their disease [91]. In patients with resectable PDAC, on the first line of treatment, (gemcitabine/nab-paclitaxel), only men displayed muscle loss [92]. While the mechanism of muscle waste in the cardiac muscle is different to that of the skeletal muscles, cardiac atrophy in cancer is also more pronounced in males than in females [93]. PDAC-derived pro-inflammatory cytokines that promote tumor growth and metastasis also activate metabolic pathways that ultimately cause skeletal muscle protein loss and cachexia [91]. Most commonly implicated pro-inflammatory cytokines in cachexia include TNF-α and interleukins IL-1, IL-6, and IL-8. IL-6 alone can induce most cachexia symptomatology, including muscle and fat wasting, the acute phase response, and anemia [94]. Different mechanisms of the development of CC between the sexes was suggested almost 40 years ago, when it was noted that female tumor-bearing mice exhibit protection from metabolic perturbations and cachexia development [95]. Findings from the *Apc ^Min/+^* mouse model of multiple intestinal neoplasia imply that, in female mice, CC is at least initially IL-6-independent [96]. In the most commonly used GEMM for PDAC—KPC mice (*LSL-Kras^G12D/+^*, *LSL-Trp53^R172H/+^*, *Pdx-1-Cre*), male mice show more severe cachexia than females [92]. In a recent study comparing the effects of KRAS:SOS1, MEK1/2, and PI3K inhibitors in the syngeneic, orthotopic, metastatic PDAC model in C57BL/6J, male and female mice were analyzed for the impact of sex on the pathological features of PDAC, its efficacy, and possible adverse side effects [97]. It was observed that not just that the key pathological features of PDAC were sex dependent, where male mice had a higher incidence of lung metastasis, more peritoneal invasion, and bigger tumors, but also that the body weight and the burrowing activity of male mice was reduced compared to the female mice, and the clinical distress score was increased. The control untreated male mice had to be euthanized much earlier due to reaching humane endpoints.

Given the sex disparities in cachexia in PDAC mouse models and in patients, a sex-segregated analysis of survival in both preclinical and clinical setting is necessary.

## 10. Pharmacokinetic Differences Between Sexes in PDAC

Pharmacokinetics in pancreatic cancer, like in other diseases, show significant differences, affecting drug efficacy and safety. Differences in drug absorption can be attributed to the different gastrointestinal physiology between men and women. Women have a lower total body water volume, extracellular and intracellular water volumes, total blood volume, red blood cell content, cardiac output, and organ blood flow rate, all of which influence the volume of distribution of a drug (Vd). In addition, in the premenstrual and late luteal phases, water retention and hyponatremia are observed, and have an impact on Vd [98]. Whenever men and women receive the same dose of a water-soluble drug, the Vd will be higher in men while the Vd of lipid-soluble drugs will be higher in women. Drug metabolism in the liver is closely dependent on cardiac output and liver blood flow, which are both lower in women than in men. Drug metabolism also depends on enzymes and transporters that exhibit sex differences in expression and activity. Many studies have shown that lower clearance rates for certain drugs in women can lead to increased exposure and potential toxicity [99,100,101]. Most chemotherapeutics are eliminated through kidneys. Drug elimination is dependent on the tubular secretion, reabsorption, and glomerular filtration rate. All these parameters are lower in women than in men [99], and the clearance of drugs that are primarily excreted through the kidneys is slower in women than in men [102].

The drugs approved for the treatment of pancreatic cancer by the Food and Drug Administration (FDA) are paclitaxel albumin, capecitabine, erlotinib, everolimus, 5-FU, gemcitabine, irinotecan, olaparib, mitomycin, and sunitinib. The current standard of care also includes drug combinations such are FOLFIRINOX (5-fluorouracil, leucovorin, irinotecan, and oxaliplatin), gemcitabine-cisplatin, gemcitabine-oxaliplatin and OFF (5-fluorouracil, leucovorin and oxaliplatin), that are used depending on the stage of the disease, the tumor subtype, the response to monotherapy, and the patient’s overall condition.

Sex differences in the pharmacokinetics and pharmacodynamics of these drugs are well documented. Yet, many major trials do not report nor analyze treatment outcomes or adverse effects by sex. Landmark trials that have set the first line chemotherapy for PDAC, for FOLFIRINOX versus gemcitabine [103], and for nab-paclitaxel plus gemcitabine versus gemcitabine alone [104,105], have not analyzed the outcome by sex. Here, we will review pharmacokinetic differences in the drug metabolism between the sexes and the outcomes from clinical trials that underline the importance of sex in response to therapy.

### 10.1. Gemcitabine

Gemcitabine (GEM), the third anticancer drug prescribed worldwide, is a deoxycytidine analog that interacts with many signaling pathways within the cell. Human nucleoside transporters (hNTs) mediate the transport of gemcitabine into the cell, after which the prodrug undergoes three-step phosphorylation processes via deoxycytidine kinase (dCK) to the monophosphate (dFdCMP) and then via pyrimidine nucleoside monophosphate kinase (UMP-CMP kinase) to gemcitabine diphosphate (dFdCDP). The production of gemcitabine triphosphate (dFdCTP), which represents the active metabolite, is still unclear, although there are indications that nucleoside diphosphate kinase may be responsible for the final step in the activation of gemcitabine [106,107,108]. A Dutch randomized PREOPANC clinical trial (NL3525) assessed the clinical outcomes in the PDAC patients undergoing neoadjuvant chemoradiotherapy (which included gemcitabine) (nCRT) versus upfront surgery for resectable and borderline-resectable pancreatic cancer [109]. In the follow-up analysis of the impact of sex on patient survival, it was found that the 5-year overall survival rate following resection preceded by nCRT was 43% for women compared to 22% for men [110]. The longer overall survival in women contributed to enhanced immunity and the prevention of the infiltration of pro-tumoral M2 macrophages into the tumor microenvironment [110].

### 10.2. Irinotecan

Irinotecan is hydrolyzed to its active metabolite, SN-38, via various polymorphic enzymes, including cellular carboxylesterases. SN-38 is inactivated via UGT1A, CYP3A4, and CYP3A5, which form several pharmacologically inactive oxidation products. The elimination of irinotecan is also dependent on the drug-transporting proteins, ABCB1 (P-glycoprotein), ABCC2 (cMOAT) and ABCG2 (BCRP), present on the bile canalicular membrane. The different processes involved in drug elimination, whether through metabolic breakdown or excretion, vary significantly between individuals [111]. Conflicting effects of sex on irinotecan pharmacokinetics have been suggested. Several studies have indicated that variability in systemic exposure to the active metabolite, SN-38, was predicted by sex. Some studies found that women may be exposed to higher concentrations of irinotecan, partly due to the reduced clearance of SN-38 [112], while others found no sex dependence [113,114,115].

### 10.3. Paclitaxel

Paclitaxel metabolism is mainly related to hepatic enzymes. In vitro studies with human liver microsomes showed that paclitaxel is oxidized by CYP2C8 to 6α-hydroxypaclitaxel (major pathway) or by CYP3A4 to p-3′-hydroxypaclitaxel (minor pathway). A retrospective population study of Joerger et al. showed that a patient’s sex significantly and independently affects paclitaxel distribution and elimination [116]. Male sex was positively correlated with the maximal elimination capacity of paclitaxel, with men having a 20% higher volume of elimination compared to women.

### 10.4. Olaparib

CYP3A4/5 enzymes primarily metabolize Olaparib. The sex differences for this specific enzyme are well documented, indicating that CYP3A4 is more expressed at the protein and mRNA level in women than in men [117,118]. The results from an animal study by Su et al. revealed that Olaparib pharmacokinetics are sex dependent, leading to high plasma exposure, long half-life, low clearance, and high bioavailability in female rats compared to male rats [119]. The sex-dependent pharmacokinetics of Olaparib can be attributed to the sex-specific expression of the CYP3A4 enzyme, which is higher in the female population [117,118].

### 10.5. Drug Combinations

The FOLFIRINOX (5-fluorouracil, leucovorin, irinotecan, and oxaliplatin) or gemcitabine and nab-paclitaxel combinations are more effective for treating metastatic pancreatic cancer (MPC) with a longer overall survival (OS) compared to the GEM monotherapy. In a study that compared FOLFIRINOX to GEM as first-line therapy in patients with MPC, 60% of the patients were male, and the authors performed survival analysis by sex and found an advantage in both the males and females for the FOLFIRINOX regimen [103]. However, almost 10 years ago a retrospective analysis of the FOLFIRINOX-treated patients with unresectable PDAC suggested that female sex was associated with a significantly higher disease control rate of 91.7% in females compared to 48.0% in male patients (*p* = 0.001), which reached 100% in female patients when primarily treated compared to treatment after surgical resection and relapse (77.8%, *p* = 0.057) [120]. A study by Kim et al. confirmed that the female sex was associated with a better response to FOLFIRINOX [121]. A total of 97 patients with MPC (54 men and 43 women) were enrolled in this study. The results suggested that females were better responders to FOLFIRINOX therapy with a trend of longer progression-free survival (PFS) (10.3 and 11.9 months) and significantly longer overall survival (OS) (17.9 and 25.9 months). During the first year of FOLFIRINOX treatment, the female patients also showed higher dose reduction in each treatment cycle compared to the male patients [121]. Results from the PRODIGE 4/ACCORD 11 randomized trial revealed that treatment with FOLFIRINOX in 342 of the MPC patients was associated with better OS (11.1 vs. 6.8 months; HR, 0.57; 95% CI, 0.45 to 0.73; *p* < 0.001) and PFS (6.4 vs. 3.3 months; HR, 0.47; 95% CI, 0.37 to 0.59; *p* < 0.001), but increased toxicity compared with GEM. Of all the measured parameters, only the hazard ratio (HR) was compared by sex and did not show any difference [122] (NCT00112658). Lambert et al. assessed the sex differences from the PRODIGE 4/ACCORD 11 trial in the FOLFIRINOX group of 171 patients (106 women and 65 men) and observed no sex/gender-related imbalance regarding demographic and clinical parameters, except for lymph nodes metastasis (17% in women and 35% in men; *p* = 0.012). Additionally, in the FOLFIRINOX group, the survival parameters were in favor of females compared to males (with OS 13.1 for women and 10.3 months for men, and PFS 7.2 for women and 5.9 months for men), but did not reach statistical significance [123]. A PRODIGE 24/ACCORD 24 multicentric randomized phase III trial recruited 493 patients with resected pancreatic ductal adenocarcinoma to compare the modified FOLFIRINOX regimen with GEM therapy. The results showed significantly longer survival in the FOLFIRINOX group compared to GEM alone. The FOLFIRINOX treatment led to a 39.7% disease-free survival rate at 3 years compared to 21.4% in the GEM group. Also, the patients in the FOLFIRINOX group had 54.4 months of median overall survival compared to 35.0 months in the GEN group, but sex differences were not considered [124] (NCT01526135).

Recently, the efficacy and safety of a new drug combination, NALIRIFOX (irinotecan, oxaliplatin, leucovorin and fluorouracil), were assessed compared to nab-paclitaxel-GEM treatment through a randomized, phase III clinical trial, NAPOLI 3. The trial included 770 MPC patients from 18 countries worldwide. The patients that received NALIRIFOX showed higher OS at 11.1 months compared to 9.2 months for the nab-paclitaxel-GEM group of patients, but no differences in the adverse events between these two groups were observed. In both of the groups, the OS was slightly in favor of the female patients (11.6 vs. 10.9 months in NALIRIFOX group 9.5 vs. 9.0 nab-paclitaxel-GEM group) [125] (NCT04083235).

In a study that compared Irinotecan plus GEM versus GEM monotherapy in patients with locally advanced or MPC, there was no survival advantage despite the increased tumor response rate in the combination group [126]. However, apart from balancing the treatment arms for sex, no sex-segregated analyses were performed for survival, tumor response, or adverse effects.

Between 1997 and 2000, 10 drugs were withdrawn from the US market. 8 of 10 were found to represent a greater health risk for women than for men, and 37% of the FDA-approved drugs between 2000 and 2002 were found to have sex differences in pharmacokinetics, efficacy, or adverse events [127]. This should serve as an urgent call to revise analysis and reporting in oncology trials as well. Efficacy and adverse effects in patients in PDAC clinical trials should be reported by sex.

## 11. Considerations, Recommendations and Measures That Can Be Taken

Disparities in the outcomes across age, sex, and race/ethnicity exist among the PDAC patients. The interplay of genetic, immune, and hormonal factors influences how PDAC unfolds and how the body responds to treatment. As most researchers do not consider sex in the study design and interpretation, mechanisms underlying sex bias in PDAC remain unrecognized. Personalized treatment for PDAC has so far been ineffective [128]. As most approaches in precision medicine assign therapy without considering sex as a variable, one has to wonder whether this was a part of the reason.

The National Institutes of Health (NIH) in the United States mandated the enrollment of women in human clinical trials in 1993, and twenty years later demanded the same in the preclinical investigation to be performed in both male and female animals [129]. In 2014, the European commission adopted similar policies. However, the mandated policies raised concerns that including both sexes would slow down research and waste resources [130]. We stand with the opposite opinion, that the costs of not taking sex into account are even higher, as they result in failed clinical trials, misdiagnosis, inappropriate therapies for women and men, and the omission of fundamental biological principles [11]. The consideration of the sex dimorphism is essential for the discovery and development of effective novel treatments for pancreatic cancer with increased efficacy and limited toxicity.

Despite recommendations, many researchers still use a disproportionately higher number of male animals. It is often assumed that results from males can be applied to females, and studies where both sexes are included frequently fail to analyze the results by sex. The lack of interest in sex differences is harmful, and also presents a missed opportunity for innovation [131]. For example, considering sex as a variable, and using RNAseq data and machine learning, Ojha et al. recently discovered unique gene expression patterns in the primary PDAC tumors in men and women, and developed a sex-specific 3-year survival predictive model that outperformed the single general model despite its smaller sample size [132]. These differences could be captured by machine learning techniques, even when the genes located on the Y-chromosome were excluded. Another great example of sex-segregated analysis that led to a significant finding is a study by Hermann et al. [88], in which PDAC liver metastasis transcriptome revealed the male-specific up-regulation of a systemic tissue inhibitor of metalloproteinases 1 (TIMP1) that was further confirmed to be a causal role for increased liver metastasis and shorter survival in males in a mouse model of the disease. This opens up the possibility for establishing a biomarker in male PDAC patients, as the levels of TIMP1 can be measured in plasma and male PDAC patients with increased TIMP1 levels, had a threefold increased risk of disease recurrence in the liver as compared with all other PDAC patients [88].

To close the sex gap in PDAC research, we propose several measures. In vitro, one should report the sex of the used PDAC cell lines and stromal cells. Female and male cells can exhibit sex differences in a transcriptional profile in culture, as well as differences in growth rate, metabolism, and response to stimuli [133]. The pancreatic cancer panel from the American Type Culture Collection (ATCC-TCP-1026) contains five male and two female cell lines. Cellosaurus has a database of more than 570 available PDAC cell lines (https://www.cellosaurus.org/, accessed on 10 October 2024), with sex reported for each one, and some analyzed phenotypically and genotypically [134].

Both females and males should be included in all the phases of preclinical and clinical drug development, and outcomes should be analyzed and reported by sex. This also applies to data from the high-throughput sequencing and genome wide association studies. Although a higher incidence of PDAC in males dictates the bias in patient recruitment, one can decide to prospectively collect or retrospectively analyze equal number of patients per sex. Sex should be used as a stratifying factor and sex-specific analyses should be performed. If differences do not exist when data are disaggregated by sex, this should also be reported. If they do exist, the source of sex difference should be determined (Figure 3). Potential sources could be genetic, epigenetic, hormonal, immune, metabolic, pharmacokinetic, environmental etc., and could lead to novel findings and ultimately the development of better-honed therapies.

Failures to translate research findings from the basic to translational domain are often attributed to issues of subjective bias, inappropriate experimental design, and statistical analysis. Studying only one sex also contributes to this failure. Some journals have recognized this issue and now demand the inclusion of sex as a biological variable [135]. At least reporting the sex of the cells, and the animals and human specimens, is encouraging, but to improve the translational benefit analytical study of both sexes is necessary. The European Union policy review on how inclusive analysis can improve research and innovation suggests to consider whether sex is a covariate, confounder, or explanatory variable [136]. Including sex into the experimental design helps achieve responsible, rigorous and reproducible science [12]. If incorporating both sexes in the research design is not possible, this should be indicated in the article titles and trial reports [137]. 

For further guidance on how to include sex as a biological variable in research, we recommend the SAGER guidelines [131] and the Gendered Innovations Annex A from the European Commission Directorate [136]. We will conclude with The European Society for Medical Oncology’s recommendation that “Men and women with nonsex-related cancers should be considered as biologically distinct groups of patients, for whom specific treatment approaches merit consideration” [138].

## Figures and Tables

**Figure 1 ijms-25-12669-f001:**
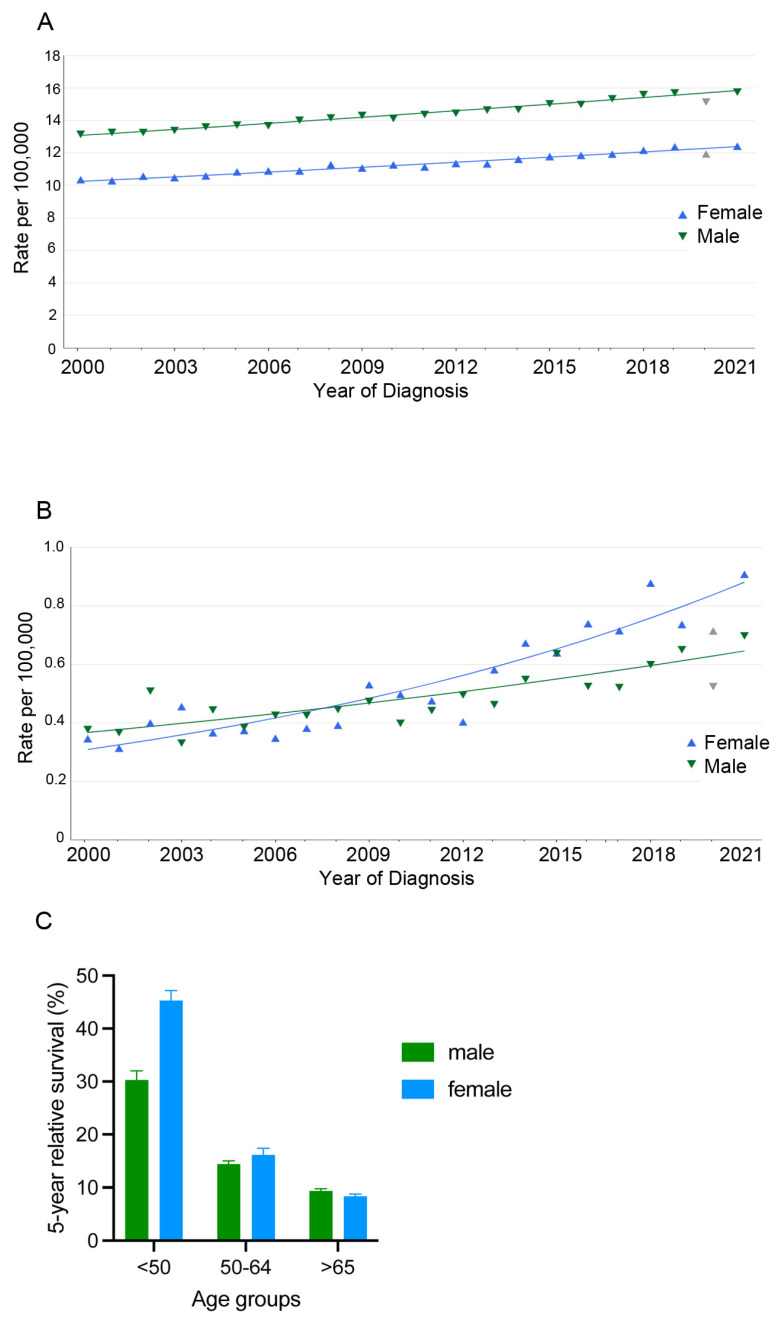
Delay-adjusted SEER incidence and 5-year relative survival rates for pancreatic cancer. (**A**) Incidence of all ethnicities, all stages, all ages; (**B**) incidence of all ethnicities, all stages, ages 15–39. (**C**) 5-year relative survival rate. Data source: Surveillance research program, National Cancer Institute. SEER incidence data; November 2023 submission; SEER, 22 registries, (https://seer.cancer.gov/registries/terms.html, accessed on 8 September 2024). The 2020 incidence rate(s) during COVID were not used in the fit of the trend line(s) and are displayed on the graph as gray data points.

**Figure 2 ijms-25-12669-f002:**
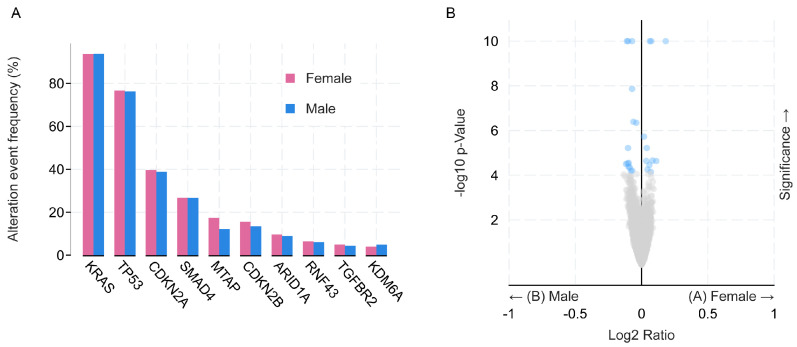
Genetic and epigenetic differences between sexes in PDAC. (**A**) Genes with highest frequency of mutation in PDAC MSK 2024 cohort; (**B**) gene methylation level in PDAC TCGA Firehouse Legacy cohort; significant difference labeled in blue (q-value below 0.05), gray circles represent genes with q-values above 0.05. Both graphs downloaded and adapted from cBioPortal, (https://www.cbioportal.org/, accessed on 10 September 2024 (**A**) and 1 October 2024 (**B**).

**Figure 3 ijms-25-12669-f003:**
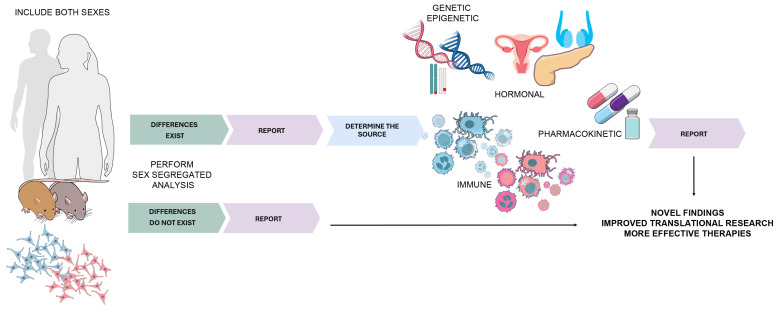
Analysis and reporting by sex. Shades of gray and brown represent two sexes, and shades of pink and blue represent female- and male-derived DNA, cells and organs, respectively. Figure generated with vectors from NIAID NIH Bioart. (https://bioart.niaid.nih.gov/vectors downloaded on 13 November 2024).

## Data Availability

The data presented in this review are publicly available in online repositories: on the BioPortal platform (https://www.cbioportal.org, accessed in September and October 2024), and on SEER*Explorer platform, an interactive website for SEER cancer statistics (https://seer.cancer.gov/statistics-network/explorer/, accessed 11 October 2024), and in the Surveillance Research Program from the National Cancer Institute, 17 April 2024 (updated: 27 June 2024; cited 11 October 2024). SEER Incidence Data, November 2023 Submission (1975–2021), SEER 22 registries.

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
