# Peer review of "Sex-Related Differences in Pancreatic Ductal Adenocarcinoma Progression and Response to Therapy"

_ijms, 2024, doi:10.3390/ijms252312669_

Round 1

Reviewer 1 Report

Comments and Suggestions for Authors

The review article "Sex-Related Differences in Pancreatic Ductal Adenocarcinoma (PDAC) Progression and Response to Therapy" by Grahovac et al. explores the influence of biological sex on various aspects of PDAC development and treatment outcomes. It discusses genetic, epigenetic, immunological, and hormonal factors that contribute to these sex-based disparities.

Major points:

The authors provide a comprehensive literature overview on sex differences in PDAC, covering genetics, immunology, and pharmacology.

Sex-based differences in cancer treatment are underexplored in PDAC, making this review relevant.

However, the review frequently cites studies that finally did not stratify results by sex. Those should be all removed, as they do not contribute to the conclusions.

Figure 3 is too trivial and should be replaced by a comprehensive graphical overview of the critical points at a glance and visualize the take-home message.

Minor points:

Page 13, line 457: 10.3. Olparib should read Olaparib.

Recommendation:

Accept with major revisions.

Author Response

We would like to thank the reviewer for the thorough examination and suggested changes to improve our review.

Major points:

  1. The authors provide a comprehensive literature overview on sex differences in PDAC, covering genetics, immunology, and pharmacology. Sex-based differences in cancer treatment are underexplored in PDAC, making this review relevant. However, the review frequently cites studies that finally did not stratify results by sex. Those should be all removed, as they do not contribute to the conclusions.

 We did cite throughout the review major studies within the PDAC research field that have marked the past decade but did not do sex segregated analysis. The point being made was that they lack this important variable, and each time we emphasized how in these important studies sex as a biological variable was not analyzed and therefore important questions remained unanswered and the results obtained may be misleading. We believe that these references need to stay in the review to, first, draw the attention on the lack of analysis in major papers in the field, and, second, invite potential researchers to revisit the data and analyze by sex. These studies are important for the conclusion of the review, as we make the statement that many researchers in the field disregard the patient or animal sex and this needs to change.

  1. Figure 3 is too trivial and should be replaced by a comprehensive graphical overview of the critical points at a glance and visualize the take-home message.

We agree with the reviewer that the figure is basic, and now have included a more comprehensive overview of the major points made in the review.

Minor points:

Page 13, line 457: 10.3. Olparib should read Olaparib.

This was a typo and has now been changed.

Reviewer 2 Report

Comments and Suggestions for Authors

Authors reviewed sex-related differences in pancreatic ductal adenocarcinoma progression and response to therapy and highlighted importance of sex as a variable during preclinical and clinical research.

Comments:

1.       Figure 1 B showed that five years relative survival rate for pancreatic cancer of male and female age group of 15-39. The rates are flipped roughly after 2007 and linearly increases the gap in relative survival rates between male and female until 2021. Is this gap linearly increasing after 2021 to till now? It is important to review research findings after the date and comment on the recent data.  

2.       Format of the figure caption: Figure 1 (a), (b), and (c) are indicated on the figure as A, B and C. These numberings should be consistent with either all lower case or upper case. Same goes for other figures throughout the review paper.

3.       Page 4 line 95: Figure 1 B is the comparative data of five years survival rate. The text states “…young females (aged 15 to 34 years [15] or 18 to 26 years [16]), compared to the males of the same age group (SEER data, Figure 1B).” Is the Figure 1B consolidate data of reference 15, 16 and SEER data? References 15 and 16 provide data of age group ranging from 15 to 34 years, but the figure has age group 15 to 39 years.

4.       Page 4 line 115 “…..the available incidence and survival data are mostly based on the data from high-income countries, as low- and middle-income countries do not have quality cancer registries.” Is this statement based on general assumption or published socioeconomic data?

5.       Page 7 line 223: What does h imply on hPDAC?  

Author Response

We would like to thank the reviewer for the thorough examination and suggested changes to improve our review.

1. Figure 1 B showed that five years relative survival rate for pancreatic cancer of male and female age group of 15-39. The rates are flipped roughly after 2007 and linearly increases the gap in relative survival rates between male and female until 2021. Is this gap linearly increasing after 2021 to till now? It is important to review research findings after the date and comment on the recent data.  

It is not possible to comment on trends and the data more recent than 2021, as the data is not yet released and publicly available. The International Agency for Research on Cancer of the World Health Organization (https://gco.iarc.fr/overtime/en) has publicly available data up to 2020.

The most recent SEER data released in April of 2024 on incidence and survival was submitted in November 2023 and covers the period from 2000 to 2021. SEER releases new data every spring and covers the data submitted by the November of the previous year.

SEER*Explorer: An interactive website for SEER cancer statistics [Internet]. Surveillance Research Program, National Cancer Institute; 2024 Apr 17. [updated: 2024 Nov 5; cited 2024 Nov 12]. Available from: https://seer.cancer.gov/statistics-network/explorer/. Data source(s): SEER Incidence Data, November 2023 Submission (1975-2021), SEER 22 registries.

  1. Format of the figure caption: Figure 1 (a), (b), and (c) are indicated on the figure as A, B and C. These numberings should be consistent with either all lower case or upper case. Same goes for other figures throughout the review paper.

We thank the reviewer for the thorough examination and we have now changed the captions to be uppercase throughout the manuscript.

  1. Page 4 line 95: Figure 1 B is the comparative data of five years survival rate. The text states “…young females (aged 15 to 34 years [15] or 18 to 26 years [16]), compared to the males of the same age group (SEER data, Figure 1B).” Is the Figure 1B consolidate data of reference 15, 16 and SEER data? References 15 and 16 provide data of age group ranging from 15 to 34 years, but the figure has age group 15 to 39 years.

We apologize if the representation of data in the figure 1B was a source of confusion. Graph presented in the Figure 1B represents visualization of the publicly available data from the SEER database released in 2023 with the SEER-defined age group 15 to 39 years. For the access to the Research Plus data from the SEER database researchers with the eRA Commons or an HHS account can download the raw data for personal further analysis. In the reference [15] authors for their own purposes performed post hoc analysis of trends between sexes and divided women in groups of 35 to 54 years and 15 to 34 years from the SEER data released in 2021. In the reference [16], authors generated groups aged 18-26 and 27-34 for the analysis of the SEER data released in 2022. We have used publicly available pre-defined group from the SEER data released in 2023 and that is stated in the figure legend. We have now included a more detailed explanation in the text of the manuscript with regards to the age groups from the references to avoid confusion.

  1. Page 4 line 115 “…..the available incidence and survival data are mostly based on the data from high-income countries, as low- and middle-income countries do not have quality cancer registries.” Is this statement based on general assumption or published socioeconomic data?

We apologize for not including references for this claim.

In order to produce reliable and reproducible data, population-based cancer registries need to adopt and follow rules for coding, data quality standards, and procedures developed by several international agencies. It has been noted that data from low- and middle-income countries often do not meet these quality criteria (Curado et al, Cancer Causes Control, 2009, PMID: 19112603).

In the analysis of 190 countries, only 60% had national cancer control policies, strategies, or action plans and 26% did not have any kind of cancer registry. Only 22% of low-income countries had a national cancer registry compared with 75% of high-income countries (Siddiqui & Zafar, J Glob Oncol 2018, PMID: 30085880). Furthermore, randomized clinical trials are performed mostly in high-income countries and do not match global cancer burden. Out of all phase 3 trials of anti-cancer therapies conducted worldwide between 2014 and 2017, only 8% were initiated and conducted in low- and middle-income countries (Wells et al, JAMA Oncol 2021, PMID: 33507236). Therefore, data in cancer research is heavily skewed toward high-income countries.

We now include these references into the manuscript, to guide interested readers into the origin of the claim in the line 115.

  1. Page 7 line 223: What does h imply on hPDAC?  

hPDAC pertains to hereditary PDAC. We omitted to include the abbreviation in line 211 on page 7, and now have corrected this.

Round 2

Reviewer 1 Report

Comments and Suggestions for Authors

The authors have improved their manuscript.